# Concerns and Expectations of Risk-Reducing Surgery in Women with Hereditary Breast and Ovarian Cancer Syndrome

**DOI:** 10.3390/jcm8030313

**Published:** 2019-03-05

**Authors:** Paola Modaffari, Riccardo Ponzone, Alberta Ferrari, Isabella Cipullo, Viola Liberale, Marta D’Alonzo, Furio Maggiorotto, Nicoletta Biglia

**Affiliations:** 1Academic Division of Gynaecology and Obstetrics, University of Turin School of Medicine, Largo Turati 62, 10128 Torino, Italy; modaffari.paola@gmail.com (P.M.); isabella.cipullo@edu.unito.it (I.C.); viola.liberale@gmail.com (V.L.); martadalonzo@libero.it (M.D.); 2Department of Gynaecological Oncology, Candiolo Cancer Institute, 10060 Candiolo, Italy; riccardo.ponzone@ircc.it (R.P.); furio.maggiorotto@ircc.it (F.M.); 3General Surgery III—Breast Surgery, Department of Surgical Sciences, Fondazione IRCCS Policlinico S. Matteo Foundation, University of Pavia, 27100 Pavia, Italy; A.Ferrari@smatteo.pv.it

**Keywords:** BRCA, Risk-Reducing Surgery, quality of life, Risk-Reducing Mastectomy, Risk-Reducing Salpingo-Oophorectomy

## Abstract

Hereditary Breast and Ovarian Cancer syndrome (HBOC) carriers face complex decisions, which might affect their fertility and body image. Using an anonymous 40-items questionnaire we evaluated the expectations and concerns about Risk-Reducing Surgery (RRS) in 204 carriers. Participants are well-informed about the options to manage cancer risk, and women with previous cancer are more concerned with screening failure. Satisfaction with RR Mastectomy is high, even if many carriers are unsatisfied with reconstructed breast feel and nipple-areola complex tactile sensation and those with previous breast cancer report a change in their sexual habits. The decrease of libido and vaginal dryness are the most complained symptoms after RR Salpingo-Oophorectomy. Nevertheless, most carriers would choose RRS again, due to cancer risk or screening-related stress reduction. Women who deferred RRS are more afraid of menopausal symptoms and cancer risk than those who had undergone or declined surgery. Women who declined RRS feel well-informed and trust screening procedures. In conclusion, HBOC carriers consider themselves well-informed and able to choose the best option for their condition, would choose RRS again because of cancer risk and screening-related stress reduction, and those who delay RRS face a higher preoperative level of concern and need support.

## 1. Introduction

Hereditary Breast and Ovarian Cancer (HBOC) syndrome is an inherited disorder in which specific genetic mutations (principally *BRCA1* and *BRCA2* genes) are associated with an increased risk for Breast (BC) and Ovarian Cancer (OC) [1,2]. Carriers should be counselled about their increased cancer risk and the options available for reducing it. Such options include screening for BC and OC, Risk-Reducing pharmacological treatment and Risk-Reducing Surgery (RRS): bilateral prophylactic mastectomy with breast reconstruction (RRM) and Salpingo-Oophorectomy (RRSO) with or without hysterectomy [2,3,4,5]. Even lifestyle modifications such as regular physical exercise, normal weight or use of oral contraceptive pills [2] may play a role in cancer risk reduction. Clinicians should provide complete counselling, including health consequences and possible side effects of each therapeutic option. When fully informed, carriers can choose the option that better suits their condition [2,5,6].

However, many factors can influence the carrier’s decision-making process [3,7] and “a posteriori” satisfaction with her own choice. Such factors might be psychological (as perceived cancer risk [5,8,9], anxiety in carrier with a previous BC [6] or body image perception [10,11]); related to surgical procedure (as in the case of expected [12,13] or unexpected side effects [14]); or linked to carriers’ quality of life, fertility, and sexuality [15,16,17,18,19]. Unfortunately, previously published studies are often focused on single aspects of the Risk-Reducing procedures in high-risk women [6,12,15,20,21,22,23,24] and composed of heterogeneous populations (high-risk women or BRCA carriers and women with previous breast cancer). As an example, the studies included in the review of Harmsen et al. [7] considered between 36 and 357 women, with a percentage of previous Breast Cancer between 0% and 51% and the BRCA carriers included varied between 13 and 55 women. However, even if surgical procedures or cancer follow-up are somehow comparable between women with sporadic BC and HBOC carriers or high-risk women, the diagnosis of genetic predisposition to cancer entails psychological and physical sequels that persist lifelong and involve carriers’ relatives and partners. This study aimed at describing expectations and concerns about RRS in women with HBOC syndrome, focusing on the effect of their choice on their quality of life and perceived body appearance.

## 2. Experimental Section

**Survey**. An anonymous 40-items questionnaire was designed to investigate expectations and concerns about RRS. The questionnaire included: (i) knowledge and concerns about RRS, postoperative complications and late effects; (ii) knowledge and concerns about screening procedures for BC and OC; (iii) role of their partner in the decision-making on RRS; (iv) satisfaction on given information on RRM surgical procedure, cosmetic and tactile sensation of reconstructed breasts; (v) influence of RRM on body appearance and sexual intimacy; (vi) satisfaction on given information on RRSO and menopausal symptoms and its possible treatments. Different question structures were used (yes – no, rating scale or matrix of multiple choices). Sometimes a free text was allowed in addition to the multiple choice answers to add supplementary opinions or feelings. Women were asked to skip the questions not applicable to their condition.

**Data analysis**. Results were analysed considering both carriers’ medical history and their choice on RRS (“RRSO/RRM”, “RRSO/RRM declined” or “intentioned for RRSO/RRM” which included both women scheduled for RRS and those who deferred surgery because of intended future childbearing). Within a rating scale from 1 to 10, we classified a score of 1–4 “low”, 5–7 “moderate” and ≥8 “high”. Results are presented as median (with range). The relationships between variables were assessed using the Mann–Whitney U Test or using the Kruskal–Wallis Test in the case of three categories. Fisher’s exact test was used to evaluate the association between categorical variables. We considered differences statistically significant if the *p*-value was <0.05 two tailed. The software SPSS 13.0 (SPSS, Inc., Chicago, IL, USA) was used for statistical analysis.

**Ethics**. All participant signed an Institutional Review Board-approved informed consent to participate in the study. Data were recorded anonymously. The study was conducted by the Declaration of Helsinki and the Institutional Ethical Committee of A.O.U. “Città Della Salute e Della Scienza”—A.O. “Ordine Mauriziano”—A.S.L. “Città di Torino” approved the study (N. CS2/147). 

## 3. Results

### 3.1. Study Population

A total of 145 adult women consulting in Our Family Cancer Clinics were invited to participate in this survey: 123 completed the questionnaire, with a participation rate of 84.8%. Women belonging to the “aBRCAdaBRA Onlus” (a national association gathering HBOC Syndrome carriers) were invited to this survey while attending a meeting of members: 85 out of 187 women accepted to participate to the study (participation rate 45.5%). Two hundred and eight women completed the questionnaire. The median age was 41 (18–68) years at diagnosis of HBOC syndrome and 47 (26–76) years at the time of the study with a median interval of 6 (0.5–30) years. Ninety-nine (48.5%) participants had previous cancer (Table 1). Moreover, one woman had a BC and a Hodgkin’s Lymphoma; while two healthy women had a neurinoma of the Hypoglossal nerve and Colorectal cancer respectively.

### 3.2. Concerns and Knowledge

Women considered themselves well-informed on the RRS and the screening procedures’ pros and cons for BC and OC (Table 2). Notably, healthy carriers were more informed on clinical and instrumental screening for BC and OC than women with previous BC or OC (*p*-value 0.0413 and 0.0042, respectively). The perceived knowledge about the effects of a healthy lifestyle on the reduction of cancer risk received a median score of 8/10 in both healthy carriers and those with previous BC or OC. The primary information sources on RRS in HBOC syndrome carriers were geneticists and gynaecologists (Table 2). Participants were moderately concerned about poor cosmetic surgical results of RRM and menopausal symptoms after RRSO (Table 2), but women with a previous BC or OC were more worried about a possible screening failure than healthy carriers (median score 7/10 versus 5/10, *p*-value 0.0102).

At the time of the survey, 185 women (90.7%) had a partner: in 84.7% of the cases, the relationship was reinforced (described as “feeling closer to my partner”) or not influenced by the diagnosis of HBOC syndrome. Nine women (4.5%) reported that they felt their partner absent after the diagnosis, three additional relationships (1.6%) divorced.

In 47.0% of cases, HBOC syndrome was no longer a major topic of conversations with the partner at the time of the survey; 38.9% of the couples talk about it with a positive mood, while the remaining 11.8% avoids talking or discuss it with a negative attitude. A total of 23 women (11.3%) did not answer this question.

### 3.3. Risk-Reducing Mastectomy (RRM)

Overall, 93 women (45.6%) had undergone RRM at a median age of 42 (24–68) years. Thirty-nine were healthy carries (39/115, 33.9%) and 54 were women with a previous BC (54/89, 60.7%). An additional 43 carriers were intentioned for receiving it.

Fifty-five out of 93 (59.2%) of participants chose RRM personally, while 36.6% (34/93) did it jointly with their partner. One woman decided in disagreement with her partner, and three (3.2%) did not answer this question.

Participants were highly satisfied with the information received about the surgical procedure and the possible postoperative complications. However, they were less confident with the information obtained about expected cosmetic results of RRM and changes in breast tactile sensation after surgery. Interestingly, women with a previous BC were less informed than healthy carriers on BC risk reduction after RRM (*p*-value 0.0147, Table 3). Moreover, women who declined surgery were less concerned about a possible screening failure than women who had undergone or deferred RRM (median score: 5 (range 1–10) versus “RRM” 8 (range 1–10) and “intentioned for RRM” 6 (range 1–10), respectively, *p*-value 0.0011). When asked about the fear of BC development, carriers who declined RRM gave a median score of 7 (range 1–10), while those who deferred RRM gave a median score of 9 (range 1–10), *p*-value 0.0114. On the contrary, no difference was found when considering women’s concern of poor cosmetic results of RRM (median score: “RRM declined” 3 (range 1–10), “RRM” 7 (range 1–10), and “intentioned for RRM” 6 (range 1–10), *p*-value 0.111).

Participants who had RRM were moderately satisfied with the shape of the reconstructed breasts and the feel of the reconstructed breast than those with previous BC, but they were unsatisfied with the nipple-areola complex tactile sensation (Table 3). Women with a previous BC more commonly reported a change in their frequency of sexual intercourse (*p*-value 0.0168) and a decline in their sexual quality of life (*p*-value 0.034), but their discomfort scores never exceeded the median value of 7/10.

Four healthy carriers and seven women with a previous BC judged the influence of RRM as profoundly negative (≥8/10) regarding their body appearance, the discomfort in being naked with their partner and sexual quality of life, but all of them would choose to undergo RRM again because of the risk reduction. Mostly, women declared to have been completely honest with their partner about their feelings and discomforts after RRM (Table 3). However, about one-third of the participants were not able to fully express their discomforts or avoided to talk about it because they were concerned about the partner’s feelings. Despite all of this, 82.8% of women scored ≥8/10 their overall satisfaction with RRM, (median score 10, range 1–10, mainly because of the reduction of BC risk and of screening-related stress. In hindsight, 98.9% of all participants would choose to undergo RRM again (Table 3).

### 3.4. Risk-Reducing Salpingo-Oophorectomy (RRSO)

A total of 128 (62.7%) of participants had RRSO at a median age of 44 (33–68 years): 63 (49.2%) of them had a previous BC. Forty-seven additional women have delayed the procedure, until childbearing completion (Table 1). Overall, 57.8% of women who had RRSO scored ≥8/10 their concern about developing OC (median score 8, range 1–10), 41.4% of women indicated a score of ≥8/10 for their fear of an OC screening failure (median score 7, range 1–10) and 37.5% of them attributed a score of ≥8/10 for related menopausal symptoms (median score 5, range 1–10). Women who delayed RRSO were more concerned about menopausal symptoms than women who had undergone or declined the procedure (median score 9 (range 1–10), 5 (range 1–10) and 6.5 (range 1–10), respectively, *p*-value 0.00337). Results were similar when considering the fear for OC, that was more frequently reported by women who delayed RRSO (median score “intentioned for RRSO” 8 (range 5–10), “RRSO” 7 (range 1–10) and “RRSO declined” 6 (range 1–8), *p*-value 0.00575).

The satisfaction of women having undergone RRSO with the information received was scored ≥8/10 by 87.8% of responders about cancer risk reduction (median score 10, range 1–10; Table 4), by 87.0% about the surgical procedure (median score 9, range 1–10) and by 78.9% about the possible complications (median score 9, range 1–10). Conversely, only 61.9% and 48.0% of the participants scored ≥8/10 their satisfaction with the information received about related menopausal symptoms (median score 8, range 1–10) and their possible treatments (median score 7, range 1–10). Interestingly, healthy carriers resulted in being more satisfied than those with previous BC with the information about OC risk reduction after RRSO and possible treatment of postoperative menopausal symptoms (*p*-value 0.0040 and 0.0424, respectively).

Sixty-nine women (53.9%) with RRSO were premenopausal younger than 45 years. Within this group, vaginal dryness and a decrease of libido were the most frequent symptoms with 65.2% and 53.6% of them reporting a score of ≥8/10, respectively. No difference in complains about hot flash onset, insomnia, weight gain, vaginal dryness, a decrease of libido, irritability and mood changes, was observed between women younger than 45 and the older ones. Excluding women with previous BC (38 carriers) to avoid the bias of adjuvant therapy, did not influence the results. Among the 65 healthy carriers who underwent RRSO, 22 (33.8%) reported that menopausal symptoms were tolerable and did not require any treatment (Table 4). Nine out of the 65 (14.3%) used Hormonal Replacement Therapy (HRT), a further five refused HRT because of BC fear, and nine more (14.3%) reported having been advised against HRT by their physicians. Fifteen healthy women (23.8%) chose phytoestrogen or herbal/dietary supplements, five (7.9%) took an antidepressant drug to reduce hot flashes; two more took sleeping pills (3.1%). Finally, 23 (35.4%) used vaginal moisturisers or lubricants, and eight more (12.3%) used local vaginal oestrogens.

Out of the 63 responders with a previous BC who underwent RRSO, 14 (22.2%) reported that menopausal symptoms were tolerable and did not require any treatment and 16 (25.4%) were afraid of HRT irrespective of having had BC; 11 (17.5%) took phytoestrogen or herbal/dietary supplements (Table 4). Nine women (14.2%) took antidepressant drugs to reduce hot flashes, four (6.2%) took sleeping pills, while nine women (14.2%) did not answer this question. Finally, 32 women (50.8%) used vaginal moisturisers or lubricants or local vaginal oestrogens.

Most women had no problems with discussing menopausal symptoms with their partners (73 women, 57.0%), but due to the fear of upsetting them, some had not been able to thoroughly explain their discomfort (32 women, 25.0%) or even to mention it (12 women, 9.4%). Women chose to have RRSO personally or together with their partner in 43.0% (55/128) and 53.9% (69/128) of the cases, respectively. Only four women reported a lack of partner support (3.1%). Patients’ median score on their satisfaction with their choice of undergoing RRSO was 10 (range 2–10), with 85.6% of women reporting a score of ≥8/10. In hindsight, 96.1% would choose it again because of the reduction in cancer risk. The five women (3.9%), who would reconsider their choice, were all premenopausal and complained of surgery-related menopausal symptoms strongly influencing their quality of life.

### 3.5. Women Who Declined both RRM and RRSO

Eight women (3.9%) declined both RRM and RRSO: Five were healthy carriers, the remaining three had a previous BC. They believed to be well-informed about screening for BC and OC with a median score of 9 (range 7–10) and 9 (range 6–10), respectively. They were satisfied with the information received about the pros and cons of RRS (median score for RRM 9 (range 6–10) and RRSO 8 (range 3–10)), with two women scoring the information ≤7/10 for RRM and four for RRSO.

They estimated to have received less information about the impact of a healthy lifestyle in cancer risk reduction (median score 7 (range 1–10), with two women reporting a score of <3/10). Clinicians were the primary source of information. The perceived cancer risk was scored 6 (range 1–8) with three healthy carriers reporting values as low as 1/10, 1/10 and 5/10. The fear of RRSO related menopausal symptoms, of screening failure and poor cosmetic results of RRM was scored 6.5 (range 1–10), 3.5 (range 1–8) and 4.5 (range 1–10), respectively. All the women, but one, had a partner at the time of mutation diagnosis and stated that this condition did not influence their relationship. 

## 4. Discussion

Decisions about Risk-Reducing strategies are tricky for HBOC syndrome carriers because pros and cons can change over the lifetime and can markedly affect not only carriers’ sense of self-identity, fertility and quality of life, but can also raise the risks for health conditions such as osteoporosis and heart disease. However, many other factors can influence the decision-making process of carriers. Howard et al. classified the reasons influencing women’s decision-making about Risk-Reducing strategies as (a) medical and physical (i.e., parity, menopausal status), (b) psychological (i.e., perceived cancer risk; cancer-related distress) and (c) social context factors (i.e., personal experience of cancer in the family) [25]. Additionally, Lewis et al. [5] found that the perceived cancer risk was one of the most important factors in the decision-making about RRSO and temporary increases when carriers experience a false-positive test result. Finally, Hooker et al. [6] found that carriers’ decisional satisfaction with *BRCA* testing and RRS choice lasts over time, but higher baseline anxiety in women with previous BC predicted a poorer long-term quality of life. In our survey, participants had a high level of knowledge about HBOC syndrome and the possible Risk-Reducing strategies. Those with previous cancer were more frequently afraid of a potential screening failure: this result may be related to an increase in their perceived cancer risk as reported by Lewis [5]. Women who declined RRS had a lower perceived cancer risk and were confident with screening procedures; unexpectedly, their median score about concern with cosmetic results and late RRS related side effects was lower than that of carriers who delayed RRM, though it was not statistically significant. Conversely, women of childbearing age who deferred RRS were the most worried about cancer risk, screening failure and surgery side effects. Most of the participants reported having been supported by their partner and feeling that their relationship was reinforced or not influenced by the diagnosis. Similarly to Howard [25], we were not able to clarify factors affecting the decision-making process about RRS; possibly the long time between the diagnosis and the survey may have mitigated some concerns about Risk-Reducing strategies and their life with partners and children. In any case, clinicians should take into consideration that women’s life goals may change over time and tailored counselling and support may reduce the decisional conflict, leading to long-lasting satisfaction which may increase over time.

### 4.1. Risk-Reducing Mastectomy

Our uptake rate of RRM was 45.6%, in agreement with the literature [26,27,28]. The uptake of RRM was higher in women with BC, which is consistent with Van Driel et al. who report an overall rate of RRM of 35.6% and a rate of 61.3% in the women with previous BC [28]. Additionally, in our experience, age is a determinant influencing the uptake rate of RRM: the majority of women who opt for surgery are between 35 and 45 years [25,26,27,28].

Our women were mildly satisfied with their reconstructed breast appearance and feel, but complained about a change in nipple-areola complex tactile sensation. Hagen et al. [21] found a high cosmetic satisfaction with 45% of women feeling the sub-muscular implants as a natural part of their body. However, they did not investigate the breast tactile sensation, and most of the women had received a skin-sparing mastectomy with the removal of the nipple-areola complex. Isern et al. [20] found a high cosmetic satisfaction with 83% of women reporting the surgical results to be as they had expected after the preoperative information. Brandberg et al. [12] found that cosmetic outcomes of RRM meet expectations in >70% of women, with the level of satisfaction higher for breast size and lower for breast softness and loss of sensitivity. *BRCA* carriers reported a lower level of satisfaction: the authors suppose that such patients might preoperatively underestimate possible surgical problems because they were committed to reducing cancer risk. Bresser et al. [14] found that unsatisfied patients perceived a lack of information about the surgical procedure (RRM and reconstruction with sub-muscular implanted silicone prostheses) and have experienced more complications, alteration of femininity, feeling the reconstructed breasts not “like their own” and still complaining of the long-term side effects. In our study, about one-third of the responders reported that their intimate relationship was influenced by RRM: 35.5% of women scored ≥8/10 their discomfort in being naked with their partner after surgery, and 29.0% reported that the procedure had an adverse influence on their sexuality. Women with previous BC were less satisfied with the feeling of the reconstructed breasts and reported a greater change in their sexuality, particularly in the frequency of intercourse and quality of sexual life. A possible explanation could be the side-effects of adjuvant therapy (chemotherapy, radiotherapy and antiestrogens) which can interfere with cosmetic results and sexuality. Brandberg et al. [15] found that 46% of 90 women who underwent RRM experienced a negative effect on an intimate situation and body self-confidence, with a significant decrease in pleasure at 1-year follow-up, even if they did not find any difference over time on habits, discomfort, or activity. Van Oostrom et al. [11] found that *BRCA* carriers, who underwent RRS, were less satisfied with their general and breast-related body image at 5 years follow-up with 56% of them reporting a lack in breast sensitivity and 69% of them a change in their sexual relationship. Finally, Den Heijer et al. [10] found that a higher preoperative general body image score (“more problems”) was a significant predictive factor for poor long-term body image in women undergoing RRM, while active coping and social support resulted as predictive for lower scores (“less problems”). Thus, the authors suggest that the exploration of these issues before RRM could help identify vulnerable women, who may benefit from additional support [10].

Even when RRM significantly changes body image and sexuality [10,11,15,20,21], women report a higher degree of satisfaction because of cancer risk and follow-up related stress reduction, as in our series. Van Oostrom et al. [11] found that healthy high-risk women were more anxious and depressed from 1 to 5 years follow-up, with a higher level of discomfort in *BRCA* carriers. However, *BRCA* carriers who opted for RRS had a lower level of anxiety and depression and estimated that RRS was worth the adverse consequences on body image and sexual relationships [11].

In our survey, women with previous BC were less satisfied with information about cancer risk reduction after RRM than healthy carriers: a possible explanation could be that they are more focused on the fear of progression of the previous BC than on the development of new cancer. Interestingly, women who deferred RRM turned out to be more concerned about interval Breast Cancer, screening failure, and poor cosmetic results than women who already received or declined RRM, corroborating the need of an attentive and tailored preoperative support.

### 4.2. Risk-Reducing Salpingo-Oophorectomy

Our uptake rate of RRSO was 67.0% in a median time since disclosure of 6 years and a median age of 44 years, in agreement with the literature: 75% at 10 years follow-up in Denmark [27], 45% at 2 years in UK [26] and 52% at 7 months in Korea [29]. Evans et al. [26] found that the uptake rate of RRSO varied between *BRCA1* and *BRCA2* carriers (52% and 28% 2 years after the diagnosis, respectively) and continued to increase over the time (predicted uptake 66% and 42% at 7 years, respectively). This difference may be the result of a “counselled risk”, based on the lower incidence and the slightly better prognosis of ovarian cancer in *BRCA2* mutation carriers. However, a recent Cochrane review shows that *BRCA* carriers who received RRSO had longer overall survival, lower BC and OC mortality and better quality of life in term of OC risk perception [30]. Meta-analyses showed that RRSO might reduce the risk of death from OC and BC in *BRCA1* carriers, but its role in *BRCA2* carriers was uncertain. However, the reliability of the evidence is very low, and the authors could not draw definitive conclusions. Mai et al. [13] found that concerns about menopausal symptoms and loss of fertility, and the perception that “surveillance was less invasive and radical than surgery,” were associated with a higher likelihood of choosing OC surveillance.

Conversely, RRSO was associated with older age, *BRCA* status, awareness of OC surveillance limitations, knowledge of the intervention-related risk and benefits, a higher perceived OC risk and related worries. Madalinska et al. [9] reported that women who underwent RRSO were less worried about their OC risk or “cancer risk among their family members” and that cancer worries had less frequently affected their mood and functioning than women who opted for surveillance. Finally, women who had RRSO would choose surgery again and would recommend it in 86% and 63% of the cases, respectively. In our survey, the perceived reduction in cancer risk is the main reason for the high level of satisfaction with RRSO as well. This result holds on even when RRSO profoundly worsened menopausal symptoms and quality of sexual life. Moreover, women who delayed RRSO experienced a higher level of concern about OC and menopausal symptoms than those who had already undergone or declined RRSO [3,7,13]. Their perceived cancer risk may be increased both because of social context components (such as establishing and maintaining couple relationships, the potential loss of ability to bear children, or concern about having cancer and leaving their children motherless [16]) and the delay in RRS.

Vaginal dryness and the decrease of libido were the most frequently complained post-surgical symptoms among our women, in line with the literature [7,9,17,31]. The hypoestrogenic condition, particularly in premenopausal women, is responsible for these symptoms and is associated with a deterioration in women’s perceived body image, quality of life and sexual health, eventually leading to emotional and psychological disorders and a loss of intimacy with the partner [17,31]. Finch et al. [22] reported a significant decline in sexual functioning and a worsening in vasomotor symptoms in 75 premenopausal women submitted to RRSO. Women starting HRT at follow-up experienced less discomfort in sexual functioning than women who did not, but never achieved the pre-menopausal comfort levels. Vermeulen et al. [23] evaluated the effect of HRT prescribed immediately after RRSO. They found that HRT-users exhibited fewer short- and long-term endocrine symptoms than HRT-non-users, with no difference from premenopausal women, who chose surveillance. Similar results were found among sexually active women when considering the effect of HRT on sexual discomfort. Thus, the authors hypothesised that prescribing HRT immediately after surgery could prevent the onset of symptoms or treat them before becoming severe. Finally, Johansen et al. [19] found that sexually active women were younger, HRT current users, had better body image, higher quality of life and health condition, and received more care from their partner than sexually inactive women. Moreover, the current use of HRT and more care from the partner were associated with increased sexual pleasure and decreased sexual discomfort. Conversely, older age, history of BC and “increased anxiety score” were associated with lower odds of being sexually active. These results highlight the complexity of women’s sexual functioning, which is also influenced by factors such as satisfaction with relationship or body image. Recently Nebgen et al. have found that healthy pre-menopausal carriers benefited from bilateral salpingectomy with delayed oophorectomy [8] in terms of cancer worry and anxiety. This approach might be a new option for those women concerned about premature menopause, loss of fertility or a decline in sexual function, but more studies are necessary.

In our survey, one-third of the healthy participants said that menopausal symptoms were tolerable and did not need any treatment. Of the remaining two thirds, 14.3% only accepted HRT, while about 85% preferred phytoestrogens or herbal/dietary supplements, antidepressant drugs and vaginal moisturisers or local vaginal oestrogens. The latter made their choice because of many reasons: a satisfactory response to non-hormonal treatment, concern about cancer risk or because advised against HRT by their physicians. The longer median interval between RRSO and the survey might explain our results as menopausal symptoms become less disturbing with passing the time.

### 4.3. Study Limitations

This study has some limitations. First, the survey was carried out using a not-validated questionnaire created after a review of the current literature. We are aware that the questions may not be able to fully describe or highlight the difference among carriers. However, differently from other authors [6,22,23,24], the questions were straight forward and aimed at recording well-defined items at the same time. Other studies did not use a validated questionnaire [16,32] or added extra questions to the validated test [9].

The questionnaire structure was a useful tool to standardise women’s answers, but the survey probably did not let participants fully express their opinions, even when some free text was allowed to add supplementary feelings or opinions. Moreover, our aim was not verifying the real scientific competence of each participant on HBOC syndrome, but understanding if and how carriers’ experience of surveillance or RRS matched the given information. We believe that this could help clinicians involved in carriers’ counselling and follow up. On the other hand, participants regularly attend our Family Clinics or participate to “aBRCAdaBRA Onlus” Association initiatives; this can explain the high level of knowledge about HBOC syndrome and the different options to manage it.

Invited women had different medical histories which could have influenced their satisfaction with RRS or concerns about cancer, but such a heterogeneous population can be found in other papers [6,7,13]. Moreover, the number of participants for each group could have influenced the survey’s ability to highlight significant differences. However, the study population was entirely made of mutation carriers, differently from others [9,20,23,24].

Finally, the long period between the genetic test the RRS and the survey could be a limit as it could lead to higher rates of satisfaction and lower levels of perceived fear and anxiety, as supposed by Den Heijer et al. [10].

## 5. Conclusions

This survey may provide useful information for improving the counselling of women with HBOC Syndrome. Carriers are satisfied with the information received on the different options to manage the HBOC related cancer risks. When they choose Risk-Reducing Surgery, their satisfaction lasts over time. However, factors such as medical history or maternal desire might influence their perception of cancer risk and risk-reducing options’ pros and cons. Clinicians should help carriers to set realistic expectations on Risk-Reducing Surgery and possible surgical-related side effects (as menopausal symptoms or nipple-areola complex tactile sensation), highlight the availability of effective endocrine and non-endocrine treatments for menopausal symptoms and provide support during the preoperative time to reduce young carriers’ concerns about screening failure and cancer risk. Clinicians play an essential role in helping High-Risk women to consider all the relevant factors during their decision-making process about risk-reducing options, including awkward issues like sexuality, fear of cancer, maternal desire or relationship.

## Figures and Tables

**Table 1 jcm-08-00313-t001:** Details on Risk-Reducing Surgery and medical history of women with Hereditary Breast and Ovarian Cancer (HBCO) syndrome included in the study.

Characteristics	Results
**Median age at survey filling-out**	47 (26–76) years
**Median age at the HBOC* syndrome diagnosis**	41 (18–68) years
**The indication of the genetic test:**	
At least one relative with a proven genetic mutation	44 (21.6%)
Previous Breast Cancer	87 (42.6%)
Previous Ovarian Cancer	11 (5.4%)
Previous Breast and Ovarian Cancer	2 (1.0%)
Multiple cases of Breast and Ovarian Cancer in the patient’s family	60 (29.4%)
**Genetic mutation**	
*BRCA1*	106 (51.9%)
*BRCA2*	84 (41.2%)
*BRCA1 + BRCA2*	2 (1.0%)
High Familial risk	12 (5.9%)
**Risk-Reducing Mastectomy (115 women, excluding 89 women with a previous BC°)**	
RRM	39 (33.9%)
Intentioned for RRM	35 (30.4%)
RRM Declined	34 (29.6%)
Not reported	7 (6.1%)
**Risk-Reducing Salpingo-Oophorectomy (191 women, excluding 13 women with a previous OC^)**	
RRSO	128 (67.0%)
Intentioned for RRSO	47 (24.6%)
RRSO Declined	10 (5.2%)
Not reported	6 (3.1%)
**Carriers who declined both RR Mastectomy and Salpingo-Oophorectomy**	8 (3.9%)

* HBOC = Hereditary Breast and Ovarian Cancer syndrome; ° BC = Breast Cancer; ^ OC = Ovarian Cancer.

**Table 2 jcm-08-00313-t002:** Results of carriers’ survey on knowledge and concerns on Risk-Reducing Surgery and screening procedures, considering their medical history (healthy one versus those who had a previous Breast or Ovarian Cancer).

Questions	Results as a Median (Range) [% Reporting a Score of ≥8/10]			
	Overall (204 pts)	Healthy (104 pts)	Previous BC or OC (100 pts)	*p*-Value *
**On a scale of 1 (incompetent) to 10 (very competent), how do you estimate your knowledge on…?**				
RR Mastectomy	9 (1–10) [80.1%]	9 (4–10) [78.8%]	9 (1–10) [83.0%]	NS
Clinical and instrumental screening for BC	9 (2–10) [83.3%]	**9 (6–10) [88.4%]**	**9 (2–10) [78.0%]**	**0.0413**
RR Salpingo-Oophorectomy	9 (1–10) [79.9%]	9 (5–10) [80.7%]	9 (1–10) [79.0%]	NS
Clinical and instrumental screening for OC	9 (1–10) [75.9%]	**9 (3–10) [82.6%]**	**8 (1–10) [69.0%]**	**0.0042**
Healthy lifestyle in reducing BC and OC	8 (1–10) [59.8%]	8 (1–10) [59.6%]	8 (1–10) [60.0%]	NS
**On a scale of 1 (little) to 10 (very), how much are you afraid of…?**				
Possible development of BC and OC	8 (1–10) [62.3%]	8 (1–10) [61.5%]	9 (1–10) [63.0%]	NS
Failure of clinical and instrumental screening	7 (1–10) [42.6%]	**5 (1–10) [36.5%]**	**7 (1–10) [49.0%]**	**0.0102**
RRSO related menopausal symptoms	7 (1–10) [45.1%]	7 (1–10) [49.0%]	7 (1–10) [41.0%]	NS
RRM related poor cosmetic results	6 (1–10) [35.3%]	7 (1–10) [39.4%]	6 (1–10) [31.0%]	NS
**Which were your sources of information on Risk-Reducing Surgery?**				
Genetic counselling	58.3% (119 pts)	61.5% (64 pts)	55.0% (55 pts)	NS°
Gynaecological consultation	24.5% (50 pts)	19.2% (20 pts)	30.0% (30 pts)	NS°
Websites, patients’ forums, patients’ society and journals	10.7% (22 pts)	10.6% (11 pts)	11.0% (11 pts)	NS°
Family and friends	6.4% (13 pts)	8.7% (9 pts)	4.0% (4 pts)	NS°

pts = participants, RRM = Risk-Reducing Mastectomy, RRSO = Risk-Reducing Salpingo-Oophorectomy; BC = Breast Cancer, OC = Ovarian Cancer, NS = not statistically significant. * *p*-value calculated with two-tailed Mann—Whitney U test. ° *p*-value calculated with Fisher Exact test.

**Table 3 jcm-08-00313-t003:** Satisfaction and concerns among carriers, who underwent Risk-Reducing Mastectomy (RRM), considering their medical history (healthy carriers and those with a previous BC).

	Results as a Median (range) [% Reporting a Score of ≥8/10]			
Questions	Overall (93 pts)	Healthy (39 pts)	Previous BC (54 pts)	*p*- Value *
**On a scale of 1 (little) to 10 (very), how much are you satisfied with the given information on…?**				
Surgical procedure RRM	9 (1–10) [76.3%]	10 (3–10) [92.3%]	9 (1–10) [63.0%]	NS
Possible postoperative complications of RRM	9 (1–10) [66.7%]	9 (3–10) [79.5%]	9 (1–10) [55.6%]	NS
Cosmetic results of RRM	8 (1–10) [53.7%]	8 (1–10) [61.5%]	8 (3–10) [46.3%]	NS
Change in breast tactile sensation after RRM	8 (1–10) [50.5%]	8 (1–10) [61.5%]	7 (1–10) [40.7%]	NS
BC risk reduction after RRM	9 (1–10) [78.5%]	**10 (1–10) [94.9%]**	**9 (1–10) [64.8%]**	**0.0147**
**On a scale of 1 (little) to 10 (very), after RRM, how much are you satisfied with…?**				
The shape of the reconstructed breast	7 (1–10) [39.8%]	7 (1–10) [46.2%]	7 (1–10) [33.3%]	NS
The nipple-areola complex tactile sensation	1 (1–10) [9.7%]	1.5 (1–10) [15.4%]	1 (1–10) [5.6%]	NS
The feel of the reconstructed breast	4 (1–10) [14.0%]	4 (1–10) [20.5%]	4 (1–10) [9.3%]	NS
**On a scale of 1 (little) to 10 (very), after RRM, can you feel a change in…?**				
Your perceived body appearance	7 (1–10) [31.9%]	5 (1–10) [33.3%]	7 (1–10) [29.6%]	NS
Your discomfort in being naked with your partner	6 (1–10) [35.5%]	5 (1–10) [30.8%]	7 (1–10) [37.0%]	NS
Your sexual intimacy with your partner	6 (1–10) [33.3%]	3 (1–10) [25.6%]	7 (1–10) [37.0%]	NS
The frequency of intercourse	5 (1–10) [29.0%]	**2 (1–10) [17.9%]**	**6 (1–10) [35.2%]**	**0.0168**
The quality of your sexual life	5 (1–10) [29.0%]	**2 (1–10) [20.5%]**	**6 (1–10) [33.3%]**	**0.0340**
**Were you honest with your partner?**				
Yes, totally	61.3% (57 pts)	69.2% (27 pts)	55.6% (30 pts)	NS°
Yes, but I could not wholly express my discomfort	19.4% (18 pts)	12.8% (5 pts)	24.1% (13 pts)	NS°
No, because I didn’t want to hurt my partner’s feeling	9.7% (9 pts)	12.8% (5 pts)	7.4% (4 pts)	NS°
No, but my partner raised the issue	2.1% (2 pts)	2.6% (1 pt)	1.9% (1 pt)	NS°
Not answered	7.5% (7 pts)	2.6% (1 pt)	11.1% (6 pts)	NS°
**In hindsight, would you choose to undergo RRM again?**				
Yes, because of the reduction in BC risk.	49.5% (46 pts)	56.4% (22 pts)	44.4% (24 pts)	NS°
Yes, because of the reduction in screening-related stress.	7.6% (7 pts)	7.7% (3 pts)	7.4% (4 pts)	NS°
Yes, because of the reduction in BC risk and screening-related stress	37.6% (35 pts)	35.9% (14 pts)	38.9% (21 pts)	NS°
No, because of the cosmetic result of RRM on body appearance	1.1% (1 pt)	-	1.9% (1 pt)	-
Not answer	4.3% (4 pts)	-	7.4% (4 pts)	-
**Overall satisfaction of RRM**	10 (1–10) [82.8%]	10 (4–10) [84.6%]	10 (1–10) [79.6%]	NS°

pts = participants, RRM = Risk-Reducing Mastectomy, BC = Breast Cancer, OC = Ovarian Cancer, NS = not statistically significant. * *p*-value calculated with two-tailed Mann—Whitney U test. ° *p*-value calculated with Fisher Exact test.

**Table 4 jcm-08-00313-t004:** Satisfaction, concerns, menopausal symptoms and related treatment among carriers who underwent Risk-Reducing Salpingo-Oophorectomy (RRSO) considering their medical history (healthy carriers and carriers with a previous BC).

	Results as a Median (range) [% Reporting a Score of ≥8/10]			
Questions	Overall (128 pts)	Healthy (65 pts)	Previous BC (63 pts)	*p*- Value *
**On a scale of 1 (little) to 10 (very), how much are you satisfied with the given information on…?**				
Surgical procedure RRSO	9 (1–10) [87.0%]	10 (1–10) [84.6%]	9 (2–10) [82.5%]	NS
Possible postoperative complications of RRSO	9 (1–10) [78.9%]	10 (1–10) [76.9%]	9 (1–10) [74.6%]	NS
Possible menopausal symptoms after RRSO	8 (1–10) [61.9%]	10 (1–10) [63.1%]	8 (1–10) [55.6%]	NS
Possible treatment for menopausal symptoms	7 (1–10) [48.0%]	**8 (1–10) [56.9%]**	**7 (1–10) [34.9%]**	**0.0424**
OC risk reduction after RRSO	10 (1–10) [87.8%]	**10 (1–10) [89.2%]**	**9 (1–10) [79.4%]**	**0.0040**
**Were you in menopause when you had RRSO?**				
Yes	29.3% (36 pts)	23.1% (15 pts)	33.3% (21 pts)	NS°
No	57.8% (74 pts)	52.3% (34 pts)	63.5% (40 pts)	NS°
Not Reported	14.6% (18 pts)	**24.6% (16 pts)**	**3.2% (2 pts)**	**0.006°**
**On a scale of 1 (little) to 10 (very), after RRSO, did you notice the appearance of…?**				
Hot flash	7 (1–10) [46.9%]	6 (1–10) [47.7%]	7 (1–10) [46.0%]	NS
Insomnia	5 (1–10) [43.8%]	5 (1–10) [41.5%]	7 (1–10) [46.0%]	NS
Weight gain	5 (1–10) [37.5%]	5 (1–8) [33.8%]	6 (1–10) [41.3%]	NS
Vaginal dryness	8 (1–10) [62.5%]	8 (1–9) [43.8%]	8 (1–10) [71.4%]	NS
Decrease of libido	8 (1–10) [55.5%]	8 (1–10) [56.9%]	8 (1–10) [54.0%]	NS
Irritability and mood changes	6 (1–10) [38.3%]	7 (1–7) [43.1%]	6 (1–10) [33.3%]	NS
Arthralgia	7 (1–10) [39.1%]	5 (1–7) [33.8%]	7 (1–10) [44.4%]	NS
**After RRSO, were menopausal symptoms tolerable?**				
Yes	28.1% (36 pts)	33.8% (22 pts)	22.2% (14 pts)	**NS°**
No	71.9% (92 pts)	66.2% (43 pts)	77.8% (49 pts)	**NS°**
**Did you take Hormonal Replacement Therapy (HRT)?**				
Yes		14.3% (9 pts)	-	-
No, I refused HRT because I was afraid of cancer		7.7% (5 pts)	-	-
No, my physician advised me against HRT		14.3% (9 pts)	-	-
**Did you use any other treatment to improve your menopausal symptoms?**				
Yes, phytoestrogen or herbal/dietary supplements	20.3% (26 pts)	23.8% (15 pts)	17.5% (11 pts)	NS°
Yes, an antidepressant drug	11.0% (14 pts)	7.9% (5 pts)	14.2% (9 pts)	NS°
Yes, sleeping pills	4.7% (6 pts)	3.1% (2 pts)	6.2% (4 pts)	NS°
No answer	7% (9 pts)	-	14.2% (9 pts)	-
**Did you use any treatment to improve vaginal dryness?**				
Yes, vaginal moisturisers or lubricants	39.8% (51 pts)	35.4% (23 pts)	44.4% (28 pts)	NS°
Yes, local vaginal oestrogens	9.4% (12 pts)	12.3% (8 pts)	6.2% (4 pts)	NS°
**In hindsight, would you choose to undergo RRSO again?**				
Yes, because of the reduction in OC risk.	96.1% (123 pts)	95.4% (62 pts)	96.8% (61 pts)	NS°
No, because of the surgical related side effects	3.9% (5 pts)	4.6% (3 pts)	3.2% (2 pts)	NS°
**Overall satisfaction of RRM**	10 (2–10) [85.6%]	10 (5–10) [90.8%]	10 (2–10) [81.0%]	NS°

pts = participants, RRSO = Risk-Reducing Salpingo-Oophorectomy; BC = Breast Cancer, OC = Ovarian Cancer, NS = not statistically significant. * *p*-value calculated with two-tailed Mann—Whitney U test. ° *p*-value calculated with Fisher Exact test.

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
