# Peer review of "Concerns and Expectations of Risk-Reducing Surgery in Women with Hereditary Breast and Ovarian Cancer Syndrome"

_jcm, 2019, doi:10.3390/jcm8030313_

Reviewer 1 Report

The highlighted results in the abstract are never given in the manuscript in table form. Why? Tables 2-4 should include a third column which summarizes the results for all women (the other two columns combined). It is unclear what the main point of the paper is. Is it that women who delay RRS experience higher levels of concern and need support? That there is some dissatisfaction but overall women choose RRS to reduce cancer risk and screening-related stress? That these women need to set more realistic expectations of RRS? The goals aren't clear and there is a lot of slicing and dicing of the results by way of subgroups and comparisons. It is hard to sift through.

There is a substantial amount of reference to previous studies in the discussion, but not in the background. In what way is this study novel compared to other studies?

More needs to be given regarding the definition of HBOC.

There are some statements about prior cancer and increased cancer risk. This should be specified in every instance as to which anatomic site of cancer.

How many women were invited to participate? What was the participation rate?  Could this be a biased responder only sample?

The questionnaire was designed specifically for this study and is not validated. This should be mentioned in the limitations. Can the full questionnaire be provided as supplementary material? What did other studies use for assessment and why was an unvalidated, newly-designed questionnaire chosen for this study?

In some cases an answer value close to 10 has negative connotation and in some cases positive. This is unclear throughout.

Were numeric answers normally distributed? If not, nonparametric tests such as Wilcoxon two-sample test and Kruskal Wallis should be used.

In Table 3, some p-values are based on the Fisher's Exact test and other p-values are based on the T-test. But the choice isn't due to the outcome being categorical or numeric since there are results of both outcomes for most questions. It seems the choice is being made based on statistical significance which is not appropriate. One choice of test should be used for the table.

I don't think an alpha level of statistical significance of 0.05 is appropriate with so many comparisons. There are 40 items on the questionnaire so a more appropriate level would be 0.001.

One section of Table 3 is misaligned ("After RRM, how much are you satisfied with...").

Section 2.4 Risk-Reducing Salpingo-Oophorectomy (RRSO) should have an associated table of results. It is difficult to get through all results with comprehension.

Why are there no p-values for Table 4?

In general, "significantly" should not be use unless it is paired with "statistically".

Author Response

The highlighted results in the abstract are never given in the manuscript in table form. Why?

Answer: Since in JCM Authors guideline the limit for abstract length is 200 words, we had to summarise our findings without reporting all the values, which are instead separately presented both in Tables 1 and 2 and along the text. However, following your suggestion, we have verified the abstract and changed some sentences in the hope of improving it.

Tables 2-4 should include a third column which summarizes the results for all women (the other two columns combined).

Answer: we did it

It is unclear what the main point of the paper is. Is it that women who delay RRS experience higher levels of concern and need support? That there is some dissatisfaction but overall women choose RRS to reduce cancer risk and screening-related stress? That these women need to set more realistic expectations of RRS? The goals aren't clear and there is a lot of slicing and dicing of the results by way of subgroups and comparisons. It is hard to sift through.

Answer: the key point of our survey was to describe women’s satisfaction and concerns about HBOC diagnosis and risk-reducing options to manage cancer risk in order to improve and tailor the counselling of BRCA carriers. Thus, when analysing our data, we tried to understand if there were any factors influencing women experience (i.e. medical history, partner’s support or maternal desire). As you pointed out, we found that women who delay RRS, experience higher levels of concern and need support, so clinicians should take it into account when counselling and following up young HBOC carriers. We have mentioned it in the text.

Even if women who chose RRS experience less concern about cancer risk and surveillance, our intent is not to affirm that RRS is the best solution. Indeed, we also highlighted that those women who intentionally chose not to undergo RRS, report low level of concern and are highly satisfied with their knowledge and experience with surveillance. Women who had chosen to undergo RRM reported a lower than expected satisfaction with reconstructed breast feel and nipple-areola tactile sensation. Other Authors raised the question about carriers’ preoperative expectation [1–4], but only Brandberg et al. [1] investigated carriers’ satisfaction with reconstructed breast sensitivity. Our conclusions are therefore intended to stress out the role of clinicians in helping BRCA carriers to consider as many factors as possible during their  decision-making process about risk-reducing options, including awkward issues like sexuality, maternal desire or relationship.

There is a substantial amount of reference to previous studies in the discussion, but not in the background. In what way is this study novel compared to other studies?

Answer:  in the literature there are many studies about single aspects of the High-risk Woman and Risk reducing procedures [5–10,3,1,11,12]. we designed this survey aiming at describing at the same time as many aspects as possible among those considered by other Authors. Participants were therefore invited to share their expectations and concerns not only about Risk-Reducing Mastectomy (as reported in other studies [2,3,5,6])  or Risk-Reducing Salpingo-Oophorectomy (as reported in [7–11]), but also on their relationship, their trust on surveillance, their body image perception. Moreover, our study population seems to be one of the largest available on this subject. The studies included in the review of Harmsen et al. [12], had considered between 36 and 357 women with a percentage of previous Breast Cancer between 0% and 51%. Moreover, the BRCA carriers included in those studies varied between 13 and 55 women. In our study, 192 BRCA carriers and 12 High Familial Risk women participated in the survey with an overall 49% of women with a previous Breast and/or Ovarian Cancer. Thus, we believe that physicians who counsel BRCA carriers might be interested in reading about concerns and expectations of this homogeneous population, which represents the population consulting every day in Family Cancer Clinics.

We changed the Introduction section accordingly, to improve it.

More needs to be given regarding the definition of HBOC.

Answer: we added a description of HBOC syndrome at the beginning of the introduction

There are some statements about prior cancer and increased cancer risk. This should be specified in every instance as to which anatomic site of cancer.

Answer: we revised the text and specified which kind of cancer was considered.

How many women were invited to participate? What was the participation rate?  Could this be a biased responder only sample?

Among carriers consulting to Our Family Cancer Clinics, 145 women were invited, and 123 completed the questionnaire, with a participation rate was 84.8%. Women belonging to “aBRCAdaBRA Onlus” were invited to this survey while attending the annual meeting of members: 85 out of 187 women accepted to participate to the survey

The questionnaire was designed specifically for this study and is not validated. This should be mentioned in the limitations.

Answer: We added it as a limitation in the appropriate Discussion Section.

Can the full questionnaire be provided as supplementary material?

Answer: Yes, a copy of the online version of the survey can be found at:

https://www.survio.com/survey/d/V6G3I9M9N5K1I8Z1Q

Unfortunately, it is written in Italian.

What did other studies use for assessment and why was an unvalidated, newly-designed questionnaire chosen for this study?

Answer: As said before, this is a survey aimed at describing several issues identified throughout a literature review.

Hereafter are listed the scales used in some of the papers we have cited in the manuscript :

-       Den Heijer et al. [4]: Body Image Score, Impact of Event Scale and Hospital Anxiety and Depression Scale.

-       Isern et al. [3]: Hospital Anxiety and Depression Scale, Swedish Short Form 36 Health Survey Questionnaire and a modified version of Ringberg et al. concerning aesthetic satisfaction

-       Brandberg et al. [1]:  Body Image Score and Sexual Activity Questionnaire

-       Vermeulen et al. [8]: Functional Assessment of Cancer Therapy-Endocrine Sub-scale and Sexual Activity Questionnaire

-       Madalinska et al. [13]: Short Form 36 Health Survey Questionnaire, global QOL item of the European Organisation for Research and Treatment of Cancer Quality of Life Questionnaire C30, Impact of Event Scale, Functional Assessment of Cancer Therapy-Endocrine Sub-scale and Sexual Activity Questionnaire

However other studies do not use a validated questionnaire [14,15].

Moreover, as an example, Madalinska et al. [13] states: “A series of single items was used to assess the level of satisfaction with or regrets about the decision to undergo prophylactic bilateral salpingo-oophorectomy or periodic gynecologic screening”. Probably, even if Authors used validated tests, they found them not completely satisfying or too focused on specific issues.

In some cases an answer value close to 10 has negative connotation and in some cases positive. This is unclear throughout.

Answer: we made it clear reporting the full question.

Were numeric answers normally distributed? If not, nonparametric tests such as Wilcoxon two-sample test and Kruskal Wallis should be used.

Answer: As most of the answers were not normally distributed, we followed your suggestion and repeated the statistical analysis using the Mann-Whitney U Test and Kruskal Wallis.

In Table 3, some p-values are based on the Fisher's Exact test and other p-values are based on the T-test. But the choice isn't due to the outcome being categorical or numeric since there are results of both outcomes for most questions. It seems the choice is being made based on statistical significance which is not appropriate. One choice of test should be used for the table.

Answer: as for the previous question, we modified our statistical analysis

I don't think an alpha level of statistical significance of 0.05 is appropriate with so many comparisons. There are 40 items on the questionnaire, so a more appropriate level would be 0.001.

Answer: We have chosen the p-value of 0.05 because the comparisons are always performed within the same population and considering one issue at a time.

One section of Table 3 is misaligned ("After RRM, how much are you satisfied with...").

Answer: we have revised it

Section 2.4 Risk-Reducing Salpingo-Oophorectomy (RRSO) should have an associated table of results. It is difficult to get through all results with comprehension.

Answer: we modified Table 4 and added the required data

Why are there no p-values for Table 4?

Answer: we have revised and modified Table 4 adding new analysis and p-values were using the Mann-Whitney U test or Fisher’s Exact Test

In general, "significantly" should not be use unless it is paired with "statistically"

Answer: we amended it

REFERENCE

1.        Brandberg, Y.; Arver, B.; Johansson, H.; Wickman, M.; Sandelin, K.; Liljegren, A. Less correspondence between expectations before and cosmetic results after risk-reducing mastectomy in women who are mutation carriers: A prospective study. Eur. J. Surg. Oncol. 2012, 38, 38–43, doi:10.1016/j.ejso.2011.10.010.

2.        Bresser, P. J. C.; Seynaeve, C.; Van Gool, A. R.; Brekelmans, C. T.; Meijers-Heijboer, H.; Van Geel, A. N.; Menke-Pluijmers, M. B.; Duivenvoorden, H. J.; Klijn, J. G. M.; Tibben, A. Satisfaction with prophylactic mastectomy and breast reconstruction in genetically predisposed women. Plast. Reconstr. Surg. 2006, 117, 1675–1682, doi:10.1097/01.prs.0000217383.99038.f5.

3.        Isern, A. E.; Tengrup, I.; Loman, N.; Olsson, H.; Ringberg, A. Aesthetic outcome, patient satisfaction, and health-related quality of life in women at high risk undergoing prophylactic mastectomy and immediate breast reconstruction. J. Plast. Reconstr. Aesthetic Surg. 2008, 61, 1177–1187, doi:10.1016/j.bjps.2007.08.006.

4.        Den Heijer, M.; Seynaeve, C.; Timman, R.; Duivenvoorden, H. J.; Vanheusden, K.; Tilanus-linthorst, M.; Menke-pluijmers, M. B. E. Body image and psychological distress after prophylactic mastectomy and breast reconstruction in genetically predisposed women : A prospective long-term follow-up study. Eur. J. Cancer 2011, 48, 1263–1268, doi:10.1016/j.ejca.2011.10.020.

5.        Brandberg, Y.; Sandelin, K.; Erikson, S.; Jurell, G.; Liljegren, A.; Lindblom, A.; Lindén, A.; Von Wachenfeldt, A.; Wickman, M.; Arver, B. Psychological reactions, quality of life, and body image after bilateral prophylactic mastectomy in women at high risk for breast cancer: A prospective 1-year follow-up study. J. Clin. Oncol. 2008, 26, 3943–3949, doi:10.1200/JCO.2007.13.9568.

6.        Hagen, A. I.; Mæhle, L.; Vedå, N.; Vetti, H. H.; Stormorken, A.; Ludvigsen, T.; Guntvedt, B.; Isern, A. E.; Schlichting, E.; Kleppe, G.; Bofin, A.; Gullestad, H. P.; Møller, P. Risk reducing mastectomy, breast reconstruction and patient satisfaction in Norwegian BRCA1/2 mutation carriers. Breast 2013, 23, 38–43, doi:10.1016/j.breast.2013.10.002.

7.        Finch, A.; Metcalfe, K. A.; Chiang, J. K.; Elit, L.; McLaughlin, J.; Springate, C.; Demsky, R.; Murphy, J.; Rosen, B.; Narod, S. A. The impact of prophylactic salpingo-oophorectomy on menopausal symptoms and sexual function in women who carry a BRCA mutation. Gynecol. Oncol. 2011, 121, 163–168, doi:10.1016/j.ygyno.2010.12.326.

8.        Vermeulen, R. F. M.; Beurden, M. Van; Kieffer, J. M.; Dorst, E. B. L. Van; Putten, H. W. H. M. Van Der; Aaronson, N. K. Hormone replacement therapy after risk-reducing salpingo-oophorectomy minimises endocrine and sexual problems : A prospective study. Eur. J. Cancer 2017, 84, 159–167, doi:10.1016/j.ejca.2017.07.018.

9.        Hooker, G. W.; King, L.; VanHusen, L.; Graves, K.; Peshkin, B. N.; Isaacs, C.; Taylor, K. L.; Poggi, E.; Schwartz, M. D. Long-term satisfaction and quality of life following risk reducing surgery in BRCA1/2 mutation carriers. Hered. Cancer Clin. Pract. 2014, 12, 1–8, doi:10.1186/1897-4287-12-9.

10.      Tucker, P. E.; Saunders, C.; Bulsara, M. K.; Tan, J. J. S.; Salfinger, S. G.; Green, H.; Cohen, P. A. Sexuality and quality of life in women with a prior diagnosis of breast cancer after risk-reducing salpingo-oophorectomy. Breast 2016, 30, 26–31, doi:10.1016/j.breast.2016.08.005.

11.      Madalinska, J. E.; Hollenstein, J.; Bleiker, E.; Van Beurden, M.; Valdimarsdottir, H. B.; Massuger, L. F.; Gaarenstroom, K. N.; Mourits, M. J. E.; Verheijen, R. H. M.; Van Dorst, E. B. L.; Van Der Putten, H.; Van Der Velden, K.; Boonstra, H.; Aaronson, N. K. Quality-of-life effects of prophylactic salpingo-oophorectomy versus gynecologic screening among women at increased risk of hereditary ovarian cancer. J. Clin. Oncol. 2005, 23, 6890–6898, doi:10.1200/JCO.2005.02.626.

12.      Harmsen, M. G.; Hermens, R. P. M. G.; Prins, J. B.; Hoogerbrugge, N.; de Hullu, J. A. How medical choices influence quality of life of women carrying a BRCA mutation. Crit. Rev. Oncol. Hematol. 2015, 96, 555–568, doi:10.1016/j.critrevonc.2015.07.010.

13.      Madalinska, J. B.; Hollenstein, J.; Bleiker, E.; Beurden, M. Van; Valdimarsdottir, H. B.; Massuger, L. F.; Gaarenstroom, K. N.; Mourits, M. J. E. Quality-of-Life Effects of Prophylactic Salpingo- Oophorectomy Versus Gynecologic Screening Among Women at Increased Risk of Hereditary Ovarian Cancer. J Clin Oncol 236890-6898. 2018, 23, doi:10.1200/JCO.2005.02.626.

14.      Payne, D. K.; Biggs, C.; Tran, K. N.; Borgen, P. I.; Massie, M. J. Women ’ s Regrets After Bilateral Prophylactic Mastectomy. 2000, 7, 150–154.

15.      Hoskins, L. M.; Roy, K. M.; Greene, M. H. Toward a New Understanding of Risk Perception Among Young Female BRCA1 / 2 “ Previvors .” Fam. Syst. Heal. 2012, 30, 32–46, doi:10.1037/a0027276.

Reviewer 2 Report

Modaffari P et al. Journal of Clinical Medicine

This paper addresses important questions to better inform women prior to making decisions on cancer risk reduction options in the context of HBOC diagnosis. In the current format, some confounders need to be addressed and/or discussed prior to publication.

·       The methods and write up of the paper does not distinguish between perceived knowledge and actual knowledge. This is especially concerning since a fair proportion of participants reported their sources of information to be non-medically oriented such as website, families and friends (~17% if the cohort). Although these women may perceive that they have a good knowledge of screening and surgical options for BRCA carriers, the information they rely on might not be accurate (same apply to women that have received their information through genetic counseling or gynecologists). In addition, their satisfaction with the information (table 3 section 1) might be influenced by the source where they are getting their information. The table 1 does not allow to see if there are difference in sources of information for healthy women versus those that have had a cancer. A questionnaire measuring the actual knowledge of women on those issues would have been useful in assessing this potential confounder. In absence of such data, this should be discussed as a limitation.

·       As indicated in Table 1, there are 20 women hat were included in the study but that were not carrier of a BRCA1 or BRCA2 mutation. The table lists that these women have PTEN, CHEK2 with or without RAD51D mutation and MLH1). This is a serious confounder as cancer risks associated with these genes are significantly different. For instance, CHEK2 is most likely a moderate penetrance gene with no know effect on ovarian cancer risk. RAD51 is much more strongly associated with ovarian cancer risk than breast. Lynch syndrome (MLH1) comes primarily with preoccupations for colorectal and uterine cancer risk. I would recommend to conduct the analysis not including those cases. There might also be differences in BRCA1 and BRCA2 carriers. Although this possibility is discussed, the data analysis methods does not allow to assess this confounder.

·       Little attention is paid to the impact of family history and prior cancer experiences (not only family history but also outcomes of cancers) in affected relatives. I would expect that negative experiences where a close relative dies of cancer does affect women’s decisions and perception, especially for healthy women. 

·       In table 4, it is unclear if some of the women were menopaused prior to RRSO. Is this table presenting data for a subset of women for which menopause was induced by RRSO? BC patients might have undergone menopause prior to RRSO especially if they received chemotherapy as part of their treatment. Menopause symptoms may also be influenced by other therapies (such as tamoxifen, transtuzumab) for which the side effects are well described and significant.

·       P values are missing from table 4.

·       Gene names should be in italic throughout the text.

·       Please review tables to ensure consistency in the number of decimals presented. (For example, in table 2, some values have no decimals, most have 1).

·       The abstract would benefit from a conclusion sentence.

Author Response

This paper addresses important questions to better inform women prior to making decisions on cancer risk reduction options in the context of HBOC diagnosis. In the current format, some confounders need to be addressed and/or discussed prior to publication.

The methods and write up of the paper does not distinguish between perceived knowledge and actual knowledge. This is especially concerning since a fair proportion of participants reported their sources of information to be non-medically oriented such as website, families and friends (~17% if the cohort). Although these women may perceive that they have a good knowledge of screening and surgical options for BRCA carriers, the information they rely on might not be accurate (same apply to women that have received their information through genetic counseling or gynecologists). In addition, their satisfaction with the information (table 3 section 1) might be influenced by the source where they are getting their information.

Answer: we regret to admit that the translation of the original question was inaccurate as it was focused on the source of information on Risk-Reducing Surgery in HBOC carriers and not on information on HBOC syndrome. We have modified it in both Table 1 and text. However, as it is stated in AIOM guidelines on Breast Cancer (the guidelines of the Italian Association of Medical Oncologists [1]) based on NCCN guidelines [2], each woman asking for a genetic test must receive a genetic counselling before been tested, as only a physician can prescribe the genetic test. Thus, non-medically-oriented sources of information can’t be the only source of information for HBOC carriers. Finally, “aBRCAdaBRA Onlus” is a nationwide association gathering HBOC Syndrome carriers with a scientific committee composed by medical expertise, as the co-author Dr A. Ferrari.

The table 1 does not allow to see if there are difference in sources of information for healthy women versus those that have had a cancer.

Answer: we have moved this issue to table 2 and done the statistical analysis considering carriers medical history

A questionnaire measuring the actual knowledge of women on those issues would have been useful in assessing this potential confounder. In absence of such data, this should be discussed as a limitation.

Answer: We agree that carriers’ perceived and actual knowledge might not be the same, but we were interested in comparing carriers’ experience about Risk-Reducing Surgery or surveillance to their expectation and satisfaction based on the given information they had. Thus, verifying their scientific competence about HBOC syndrome was not the intent of our survey. However, as you have suggested, we add this point among the study limitations in the Discussion Section.

·       As indicated in Table 1, there are 20 women hat were included in the study but that were not carrier of a BRCA1 or BRCA2 mutation. The table lists that these women have PTEN, CHEK2 with or without RAD51D mutation and MLH1). This is a serious confounder as cancer risks associated with these genes are significantly different. For instance, CHEK2 is most likely a moderate penetrance gene with no know effect on ovarian cancer risk. RAD51 is much more strongly associated with ovarian cancer risk than breast. Lynch syndrome (MLH1) comes primarily with preoccupations for colorectal and uterine cancer risk. I would recommend to conduct the analysis not including those cases. There might also be differences in BRCA1 and BRCA2 carriers. Although this possibility is discussed, the data analysis methods does not allow to assess this confounder.

Answer: we have checked those 20 women:

Three carriers had a pathogenetic Single Nucleotide Variant of BRCA2 gene

1 Other carriers had a pathogenetic Single Nucleotide Variant of BRCA1 gene

One carrier had MLH1 mutation

One carrier was affected by Cowden Syndrome

One carrier had both CHECK2 and RAD 51D mutation

One carrier had only CHEK2 mutation

Other 12 women had a High Familial Risk and the advice to consider RRS by geneticists.

As Suggested, we have removed the four women (with CHECK or MLH1 mutation and Cowden Syndrome) from the survey population and specified those with High Familial Risk in Table 1. Thus, the survey population is now composed of 204 women

·       Little attention is paid to the impact of family history and prior cancer experiences (not only family history but also outcomes of cancers) in affected relatives. I would expect that negative experiences where a close relative dies of cancer does affect women’s decisions and perception, especially for healthy women.  

Answer: we agree with your observation, as also reported by Howard et al. [3], and we recognise that this could be a limit to our survey. Unfortunately, due to the questionnaire’s length, we had decided to reduce the questions about social context factors in favour of investigation carriers’ opinion about motherhood (data not shown).

·       In table 4, it is unclear if some of the women were menopaused prior to RRSO. Is this table presenting data for a subset of women for which menopause was induced by RRSO? BC patients might have undergone menopause prior to RRSO especially if they received chemotherapy as part of their treatment. Menopause symptoms may also be influenced by other therapies (such as tamoxifen, transtuzumab) for which the side effects are well described and significant.

Answer: we have added the issue in table 4, however in the text was specified that: “No difference in complains about hot flash onset, insomnia, weight gain, vaginal dryness, a decrease of libido, irritability and mood changes, has been observed between women younger than 45 and the older ones. Excluding women with previous BC to avoid the bias of adjuvant therapy did not influence the results.

·       P values are missing from table 4.

Answer: we revised Table 4 adding new analysis and p-values has been calculated using the Mann-Whitney U Test or Fisher’s Exact Test, as suggested by Reviewer n 1.

·       Gene names should be in italic throughout the text.

Answer:  We did it. 

·       Please review tables to ensure consistency in the number of decimals presented. (For example, in table 2, some values have no decimals, most have 1).

 Answer: we have checked and corrected all those values which did not have a decimal.

·       The abstract would benefit from a conclusion sentence.

Answer: we have modified the conclusion of the abstract to clarify our findings.

REFERENCE

1.        Associazione Italiana di Oncologia Medica, A. Linee guida NEOPLASIE DELLA MAMMELLA.

2.        3.2019, N. C. P. G. in O. V. Genetic / Familial High-Risk Assessment : Breast and Ovarian. NCCN Clin. Pract. Guidel. Ocology Version 3.2019 2019.

3.        Howard, A.; Balneaves, L.; Bottorff, J. Women’s Decision Making about Risk-Reducing Strategies in the Context of Hereditay Breast and Ovarian Cancer: A Systematic Review. J Genet Couns. 2009, 18, 578–597, doi:10.1007/s10897-009-9245-9.

Round  2

Reviewer 1 Report

The authors responded to my previous comment regarding the novelty of their research, but the introduction was not amended to reflect the literature. I still think this would improve the manuscript.

Section 2.0 has a header: Questionnaires. However, the word ‘questionnaire’ throughout the section has been edited to ‘survey’. I would suggest keeping the reference to the instrument the same throughout for consistency. In section 2.1 it is referred to as both questionnaire and survey.

At the top of page 3, it is stated “Results are presented as median (with range) according to the distribution of the data).” This sentence needs to be edited further now that mean/st.dev wording has been removed.

On page 3 it is noted that the Kruskal Wallis test was used for 3+ categories. Was this ever actually used? If not, remove the wording.

In Table 1, the section with 115 women (Risk Reducing Mastectomy) has categories that only total to 114.

Why have some of the numbers changed, including the overall sample size?

In Tables 2-4, I suggest clarifying that it is healthy carriers instead of healthy patients and also that the p-value compares healthy carriers to women with previous BC or OC.

In Tables 3-4, the comparison of overall satisfaction at the bottom of each table needs the Fisher’s Exact test foot note symbol.

Consider adding the participation rate.

Author Response

The authors responded to my previous comment regarding the novelty of their research, but the introduction was not amended to reflect the literature. I still think this would improve the manuscript.

Answer: we have added further considerations; however we prefer not to dwell on the literature in the Introduction Section due to the length of the manuscript.

Section 2.0 has a header: Questionnaires. However, the word ‘questionnaire’ throughout the section has been edited to ‘survey’. I would suggest keeping the reference to the instrument the same throughout for consistency. In section 2.1 it is referred to as both questionnaire and survey.

Answer: we have changed the word “questionnaire” into “survey” along the text unless this was referred to the questionnaire itself.

At the top of page 3, it is stated “Results are presented as median (with range) according to the distribution of the data).” This sentence needs to be edited further now that mean/st.dev wording has been removed.

Answer: we did it

On page 3 it is noted that the Kruskal Wallis test was used for 3+ categories. Was this ever actually used? If not, remove the wording.

Answer: we applied the Kruskal Wallis test in the comparison of “RRS declined”, “RRS done” and “RRS deferred” group. Results are reported along the Results Section.

In Table 1, the section with 115 women (Risk Reducing Mastectomy) has categories that only total to 114.

Answer: we apologise for that clerical mistake, we have rectified it.

Why have some of the numbers changed, including the overall sample size?

Answer: During the first round of revision, the second Reviewer asked us to remove some participants from the study population because they were carriers of genetic mutation other than BRCA. Here we report his/her comment: “PTEN, CHEK2 with or without RAD51D mutation and MLH1). This is a serious confounder as cancer risks associated with these genes are significantly different. For instance, CHEK2 is most likely a moderate penetrance gene with no known effect on ovarian cancer risk. RAD51 is much more strongly associated with ovarian cancer risk than the breast. Lynch syndrome (MLH1) comes primarily with preoccupations for colorectal and uterine cancer risk. I would recommend conducting the analysis, not including those cases.”

Here is our answer to his/her comments: we have checked these 20 women:

Three carriers had a pathogenetic Single Nucleotide Variant of BRCA2 gene

One carrier more had a pathogenetic Single Nucleotide Variant of BRCA1 gene

One carrier had MLH1 mutation

One carrier was affected by Cowden Syndrome

One carrier had both CHECK2 and RAD 51D mutation

One carrier had only CHEK2 mutation

Other 12 women had a High Familial Risk and the advice to consider RRS by geneticists, even in the absence of a detectable mutation.

As Suggested, we have removed the four women (with CHECK or MLH1 mutation and Cowden Syndrome) from the survey population and specified those with High Familial Risk without mutation in Table 1. Thus, the survey population is now composed of 204 women

In Tables 2-4, I suggest clarifying that it is healthy carriers instead of healthy patients and also that the p-value compares healthy carriers to women with previous BC or OC.

Answer: as specified in the footnote of each table, “pts” is the contraction of “participants” and not of “patients”, as the word “participant” was used along the text to refer to those carriers who have completed the survey.  

In Tables 3-4, the comparison of overall satisfaction at the bottom of each table needs the Fisher’s Exact test foot note symbol.

Answer: we amended it

Consider adding the participation rate.

Answer: we added it at the beginning of the Study Population Section

Round  3

Reviewer 1 Report

The authors responded to my concerns.

This manuscript is a resubmission of an earlier submission. The following is a list of the peer review reports and author responses from that submission.

Round  1

Reviewer 1 Report

The discussion section is too long and not appropriate for the overall length of the paper

I do also feel that at least one paragraph about the chance of proposing salpingectomy instead of oophorectomy in those women already subjected to mastectomy should be introduced, even as a proposal for the next future 

Reviewer 2 Report

This is a study of expectations and concerns regarding risk-reduction surgery among women with hereditary breast and ovarian cancer syndrome. The authors characterize the experience of women with HBOC and present findings including that women who underwent risk reduction surgery had improved quality of life relative to those who did not. This is an important question, however there are major methodologic issues that prevent this study from adding to the existing literature.

A flow diagram of which patients were included in the results would be helpful, as I found this very confusing. 

These included groups (previous cancer and health status) are not at all homogeneous. Results should be restricted to healthy women. There is too much recall bias (and confounding) when answering a survey six (!) years after cancer diagnosis. 

Saying that RRS “improves” quality of life is incorrect in this cross-sectional (and also heavily confounded) study. Please amend all causal statements, which are not possible in this study design.

Were the women who filled out the survey online similar in characteristics to those filling it out with “aBRCAdaBRA Onlus?” Were survey results different between these two groups? If not they should be stratified as well.

It would be extremely important to state the timeframe of when this survey was filled out in relation to time of surgery. Without this the results are not interpretable.

Should not present p-values as “NS” – rather, list the value.

Table 2 is going to be completely different for women who have undergone surgery vs. those who have not. Results should be stratified as such.

Inference / P-values should be double-checked throughout. For example, in Table 3, “Possible postoperative complications of RRM,” a t-test would not give p=0.0484 based on difference of the observed means (mean 1, 8.2; mean 2, 7.7) and standard deviations (sd 1, 2.4; sd 2, 2.7).

Reviewer 3 Report

Review for Manuscript cancers-386612-peer-review-v1

General Comments: Overall, very clear study design, presentation of results, and writing in the manuscript. A few general comments, but more specific comments are listed below by section and line number.

1)    For the results, please discuss if any of these individuals in the study had another cancer type not associated with HBOC organs (breast and ovary).

2)    In Table 3 and associated Results text, compare the percentage responses for each response between “Healthy” and “Previous BC” with a Chi-Square or Fisher’s Exact test.

More Specific Comments:

Title – None

Abstract

1)    Line 19-20 – Rephrase/reword the first sentence of the abstract

2)    Line 24 – Change “of breast” to “with breast”

Introduction

1)    Line 51 – Change “women” to “their”

2)    Line 53 – Change “consequence” to “consequences”

Results

1)    Line 77 – Change “procedures” to “procedures’ ”

2)    Line 110 – Change “received information” to “information received”

3)    Line 115 – Change “to be” to “they were”

4)    Line 164 – Insert a comma before “respectively”

5)    Line 173 – Change “flashes” to “flash”

6)    Line 206-208 – Rephrase/reword the sentence starting with “The fear”

Discussion

1)    Line 240 – Change “said to be” to “were”

2)    Line 322 – Eliminate space between “BRCA” and “1”

3)    Line 337 – Insert symbol between “P” and “0.05”

4)    Line 388 – Should “practicing sport” be “exercise”?

5)    Line 402 – Change “option” to “options”

Methods

1)    Line 419 – Insert “their” before “partner”

Conclusions – None

Figure and Table Legends – None

Figures and Tables – See General Comment above.